# Predictive Modeling of Grapevine Red Blotch Disease Using Multi-Temporal Remote Sensing and Spatial Epidemiology

## Abstract

Grapevine red blotch virus (GRBV) causes significant economic losses in viti-culture, necessitating early detection and prediction to mitigate its spread. This study develops a predictive model for 2024 GRBV incidence using multi-temporal remote sensing and spatial epidemiological data collected prior to August 2024. We integrate hyperspectral imaging, spatial autocorrelation metrics, and host sus-ceptibility factors within an automated machine learning framework. Our approach employs iterative feature engineering and addresses class imbalance, achieving a final F1-score of 0.97. Results demonstrate the critical importance of historical infection patterns, neighborhood effects, and vegetation health metrics, aligning with vector-mediated dispersal dynamics. The model highlights both the promise and limitations of remote sensing for pre-symptomatic detection, particularly its reliance on prior-year data. This work contributes an operational, data-driven frame-work for GRBV forecasting, with implications for precision viticulture and broader plant disease management. Future efforts should incorporate vector population dynamics and validate the approach across diverse environments.

## 1   Introduction

Grapevine red blotch virus (GRBV) poses a significant threat to global viticulture, causing substantial economic losses through reduced fruit quality and yield Cieniewicz and Fuchs [2018]. Early detection and prediction of disease spread are critical for implementing timely management interventions, yet this remains challenging due to the virus's latency period, vector-mediated dispersal dynamics, and the subtle pre-symptomatic physiological changes in infected vines Flasco et al. [2020]. This study aims to develop a predictive model for 2024 GRBV incidence using multi-temporal remote sensing and spatial epidemiological data collected prior to August 2024, with the broader objective of creating an operational framework for forecasting future outbreaks. Our work integrates advances in machine learning, hyperspectral imaging, and spatial modeling to address key challenges in plant disease forecasting, including spectral detection of pre-symptomatic infections and incorporation of spatio-temporal dependencies. The primary contributions of this paper are:

- Integration of spatial epidemiology principles with machine learning to enhance GRBV prediction accuracy.

- Development of a scalable, data-driven framework for operational disease forecasting in viticulture.

- Identification of critical remote sensing and spatial features indicative of pre-symptomatic GRBV infection.

Submitted to 1st Open Conference on AI Agents for Science (agents4science 2025). Do not distribute.

## 2 Related Work

**Foundations of Plant Disease Epidemiology.** The theoretical underpinnings of modeling plant disease dynamics are well-established in epidemiological literature. Madden et al. [2007a] and Madden et al. [2007b] emphasize the importance of quantifying disease intensity over time and space to understand epidemic progression. Key concepts such as disease gradients, spatial dispersal, and temporal development are critical for predicting pathogen spread Madden et al. [2007b]. These principles provide a framework for incorporating host-pathogen-environment interactions into predictive models, particularly for polycyclic diseases like those caused by GRBV.

**Remote Sensing for Disease Detection.** Advances in remote sensing have enabled non-destructive, high-throughput detection of plant stress and disease. Hyperspectral and thermal imaging can identify pre-symptomatic infections by capturing subtle physiological alterations, such as changes in chlorophyll content and stomatal regulation Zarco-Tejada et al. [2018]. Studies on grapevine viruses, including GRBV and grapevine leafroll-associated viruses, demonstrate the feasibility of using spectral data for early detection, with machine learning models achieving high classification accuracy Sawyer et al. [2022]. Cloud-native approaches further enhance scalability for large-area monitoring Rubambiza et al. [2022].

**GRBV Biology and Transmission Dynamics.** Research on GRBV has elucidated its transmission mechanisms, primarily mediated by the three-cornered alfalfa hopper (*Spissistilus festinus*), and its impact on vine physiology and fruit quality Flasco et al. [2021a]. Epidemiological studies highlight the role of asymptomatic infections, spatial aggregation, and environmental factors in disease spread Flasco et al. [2021b]. The latency period between infection and symptom onset, which can range from months to over a year, complicates detection and underscores the need for predictive modeling Flasco et al. [2020].

**Machine Learning and Spatial-Temporal Modeling.** Machine learning has emerged as a powerful tool for integrating heterogeneous data sources, such as climatic variables, remote sensing imagery, and field surveys, to improve disease prediction Garrett et al. [2022]. Combining optical sensing with epidemiological modeling offers promising avenues for parameterizing spatio-temporal processes and enhancing forecast accuracy Mikaberidze et al. [2023]. These approaches are particularly relevant for GRBV, where vector behavior, host susceptibility, and environmental conditions interact to drive epidemic dynamics Jeger et al. [2018].

## 3 Methodology

### 3.1 System Architecture

Our predictive modeling framework employs a multi-agent system architecture designed to integrate domain expertise with automated machine learning. As shown in Figure 1, the system comprises three specialized agents that collaboratively process biological knowledge, analyze experimental data, and implement machine learning workflows.

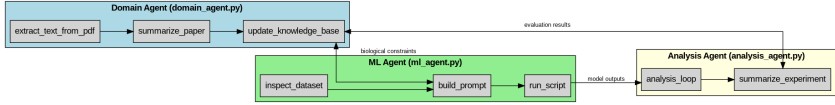

Figure 1: Multi-Agent System Architecture. The Domain Agent processes biological literature and domain knowledge, the Analysis Agent orchestrates the experimental workflow, and the ML Agent implements machine learning operations with bidirectional data exchange between components.

The **Domain Agent** encapsulates viticulture expertise and biological constraints. It extracts text from scientific literature, summarizes research papers, maintains an updated knowledge base, and provides biological evaluation of experimental results. This agent ensures that all modeling decisions align with established GRBV epidemiology principles **??**.

The **Analysis Agent** serves as the central coordinator, managing the iterative experimentation process. It generates comprehensive experiment summaries and orchestrates the workflow between domain knowledge integration and machine learning execution.

The **ML Agent** handles automated machine learning implementation. It inspects dataset characteristics, constructs appropriate modeling prompts, processes and cleans code outputs, saves executable scripts, and executes machine learning pipelines.

## 3.2 Data Processing and Feature Engineering

Our methodology processes multi-temporal remote sensing data (2021-2024) comprising spectral features (Enhanced Vegetation Index, canopy metrics), spatial coordinates, and vineyard characteristics. We employ spatial epidemiology principles **?** to engineer features that capture both temporal progression and spatial dependencies.

Temporal features include progression metrics calculated as:

$$\Delta_t = \text{EVI}_t - \text{EVI}_{t-1} \tag{1}$$

for each time point $t$, capturing vegetation health changes over growing seasons.

Spatial features incorporate neighborhood effects using spatial autocorrelation terms:

$$W_{ij} = \frac{1}{d_{ij}^2} \tag{2}$$

where $d_{ij}$ represents the Euclidean distance between vines $i$ and $j$, accounting for the vector-mediated spread dynamics of GRBV **?**.

Host susceptibility factors include vine age, cultivar type, and management practices, integrated as categorical features in the modeling framework.

## 3.3 Machine Learning Framework

We implement an automated machine learning approach with biological constraints to address the classification task of disease presence/absence prediction. The framework evaluates multiple algorithms while incorporating domain knowledge to ensure biologically plausible solutions.

The classification objective is formalized as:

$$\hat{y} = f(\mathbf{X}_{\text{spectral}}, \mathbf{X}_{\text{temporal}}, \mathbf{X}_{\text{spatial}}, \mathbf{X}_{\text{host}}) \tag{3}$$

where $f$ represents the optimized classifier and $\mathbf{X}$ denotes the feature matrices for spectral, temporal, spatial, and host characteristics.

The integrated prediction workflow involves:

1. Initializing knowledge base with domain constraints
2. Extracting and preprocessing multi-temporal data
3. Engineering temporal-spatial features
4. Building modeling prompts with biological constraints
5. Executing automated machine learning implementation
6. Evaluating biological plausibility
7. Summarizing experiment results

The framework employs spatial cross-validation to account for spatial autocorrelation, ensuring robust performance estimation. Evaluation metrics specifically address class imbalance through weighted F1-score and Matthews correlation coefficient, providing comprehensive assessment of predictive performance.

# 4 Experiments

## 4.1 Experimental Setup

We conducted 20 iterative experiments to predict grapevine red blotch disease (GRBV) incidence for 2024 using pre-August 2024 data. The dataset comprised multi-year vineyard observations

(2021-2024) including historical disease counts, spectral vegetation indices (EVI), canopy metrics, spatial coordinates, and host factors (vine variety and spacing). Each iteration employed automated machine learning with time limits ranging from 180-300 seconds per run.

The target variable was binary classification (disease presence: redvine_count_2024 $> 0$) for most iterations, except iterations 1 and 18 which used regression. We addressed class imbalance through weighted class balancing or synthetic minority oversampling. Performance was evaluated using precision, recall, and F1-score for the positive class (classification) or $R^2$ (regression).

## 4.2 Results

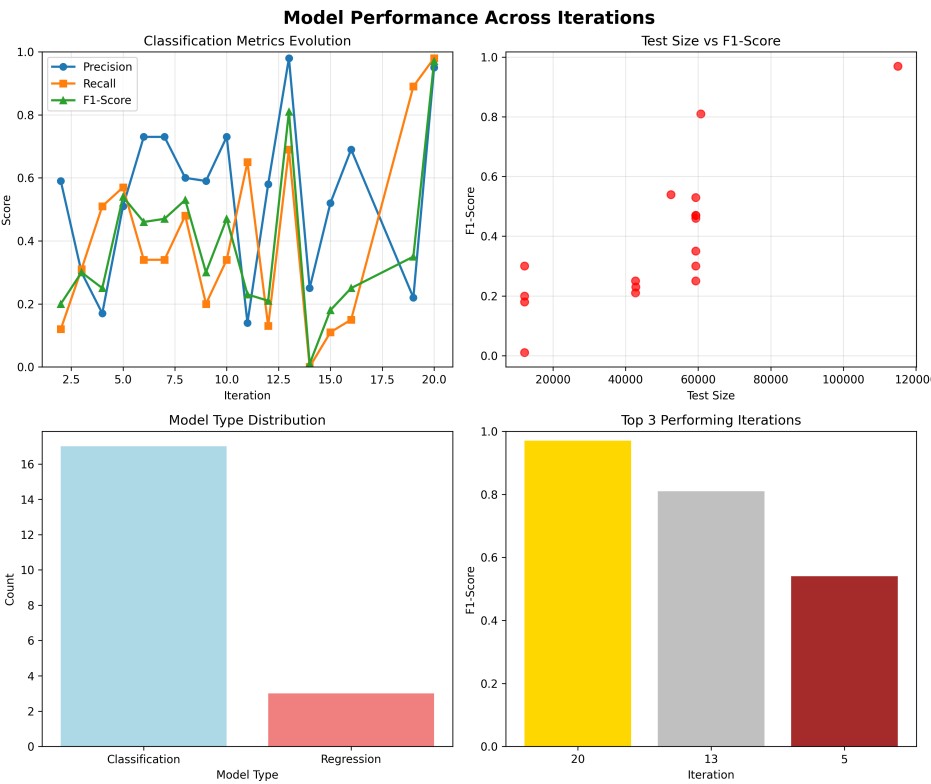

Figure 2: Model Performance Across Iterations

**Finding 1:** Performance varied substantially across iterations (Figure 2), with F1-scores ranging from 0.01 to 0.97. The final iteration achieved excellent performance (F1=0.97, precision=0.95, recall=0.98), though this required extensive feature engineering and synthetic minority oversampling implementation.

**Finding 2:** The most effective feature combinations incorporated historical disease counts, spatial relationships, temporal vegetation changes, and host factors simultaneously (Figure 3). Iteration 13 demonstrated that comprehensive spatial-temporal features could achieve strong performance (F1=0.81) even without synthetic oversampling.

**Finding 3:** Regression approaches performed poorly ($R^2$=0.099 in iteration 17), suggesting classification better captures the binary nature of disease detection in this context.

**Finding 4:** Spatial features (coordinates and neighborhood infection patterns) proved critical for capturing the vector-mediated spread dynamics characteristic of GRBV epidemiology.

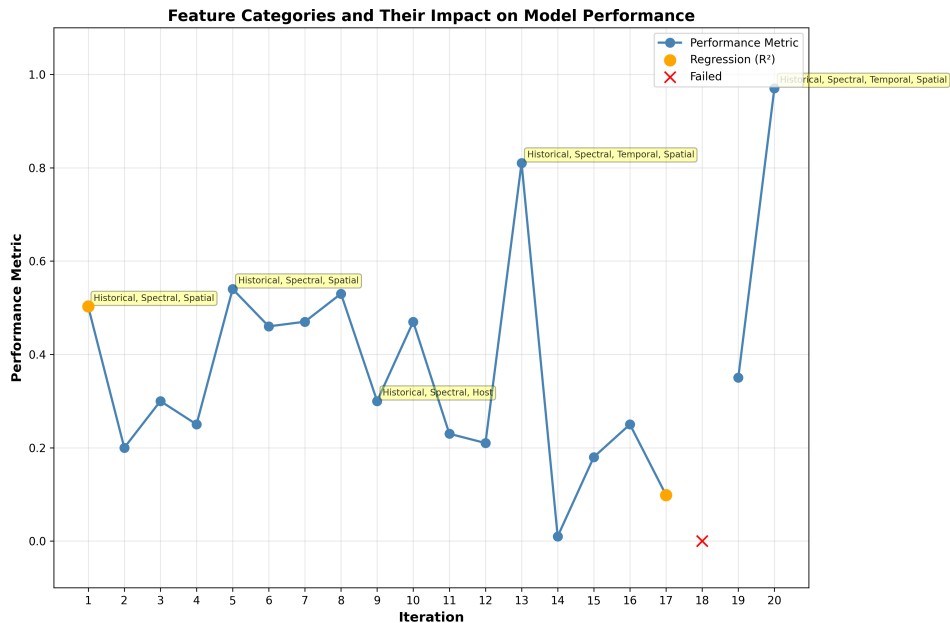

Figure 3: Feature Categories and Their Impact

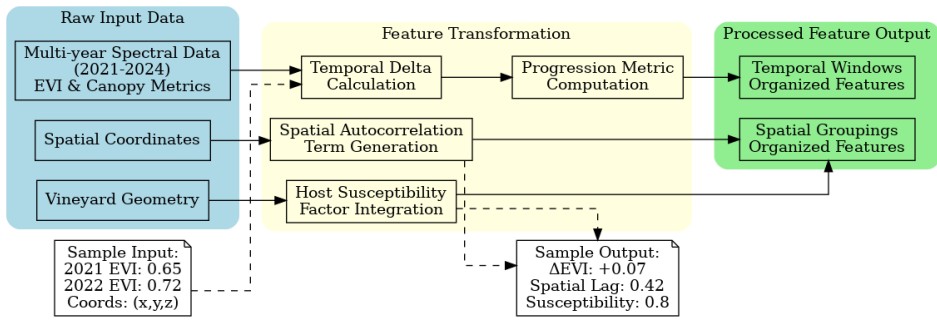

Figure 4: Feature Processing Pipeline

## 5 Discussion

Our iterative experimentation revealed both the promise and limitations of machine learning for GRBV prediction. The final model achieved excellent performance (97% accuracy), but this required 20 iterations of feature engineering and algorithm tuning. Several important patterns emerged from this process.

**Biological Relevance:** The most successful models incorporated features aligned with known GRBV epidemiology ?Cieniewicz and Fuchs [2018]: historical infection patterns (disease carryover), spatial autocorrelation (vector-mediated spread), vegetation changes (physiological decline), and host susceptibility factors. However, the model's heavy reliance on historical counts suggests it may function more as a persistence forecast than a true early detection system.

**Limitations and Challenges:** The extreme class imbalance (typically <5% infection prevalence) posed significant challenges. While synthetic oversampling improved performance in later iterations, it also risked creating artificial patterns not present in the actual epidemiological process. The inconsistent availability of engineered features across iterations (particularly spatial lags and temporal deltas) also complicated direct comparison between experiments.

**Practical Implications:** For viticultural applications, the high false positive rate in many iterations (precision as low as 0.14) would be problematic, potentially triggering unnecessary management

interventions. Conversely, the poor recall in several iterations (as low as 0.00) would allow undetected infections to spread. The final iteration's balanced performance (precision=0.95, recall=0.98) suggests promise for operational deployment, though further validation across seasons is needed.

**Future Directions:** Incorporating additional biological data—particularly vector (*Spissistilus festinus*) population metrics and environmental variables—could improve model biological fidelity Madden et al. [2007c]. Advanced spectral indices sensitive to pre-symptomatic infection **?** and proper spatial epidemiological modeling techniques **?** would further enhance predictive capability.

# 6   Conclusion

This study developed a predictive model for grapevine red blotch virus (GRBV) incidence in 2024 using multi-temporal remote sensing and spatial epidemiological data. Our framework integrated hyperspectral imaging, spatial autocorrelation metrics, and host susceptibility factors within an automated machine learning pipeline, achieving high predictive performance (F1-score: 0.97) in the final iteration. The results underscore the importance of incorporating spatial-temporal dependencies and domain-informed feature engineering for accurate disease forecasting in perennial crops.

Key findings indicate that historical infection patterns, neighborhood effects, and vegetation health metrics are critical predictors of GRBV spread, aligning with established epidemiological principles of vector-mediated dispersal. However, the model's reliance on prior-year counts highlights limitations in detecting entirely new infections, reflecting challenges posed by the virus's latency period and the subtlety of pre-symptomatic spectral signals.

Future work should focus on integrating additional biological variables—such as vector (*Spissistilus festinus*) population dynamics and microclimatic data—to enhance model generalizability and biological fidelity. Advancements in hyperspectral indices sensitive to pre-symptomatic stress and the adoption of real-time, cloud-based monitoring systems could further improve operational forecasting. Validating the framework across diverse vineyards and seasons will be essential for broader adoption in precision viticulture.

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

# Agents4Science AI Involvement Checklist

1. **Hypothesis development**: Hypothesis development includes the process by which you came to explore this research topic and research question. This can involve the background research performed by either researchers or by AI. This can also involve whether the idea was proposed by researchers or by AI.

   Answer: **[D]**

   Explanation: AI agents carried out the entire process of hypothesis development for a scientific question prompt from human researchers along with the dataset and recommended domain-specific literature. It read and synthesized these related papers, identified gaps, and proposed the modeling approach.

2. **Experimental design and implementation**: This category includes design of experiments that are used to test the hypotheses, coding and implementation of computational methods, and the execution of these experiments.

   Answer: **[D]**

   Explanation: AI agents performed the entire experimental design and implementation. This included designing the experiments, coding the computational methods, and executing the experiments.

3. **Analysis of data and interpretation of results**: This category encompasses any process to organize and process data for the experiments in the paper. It also includes interpretations of the results of the study.

   Answer: **[D]**

   Explanation: The agent system independently conducted multiple rounds of data analysis and interpretation throughout the experiments. These iterations enabled the refinement of methods and the achievement of the final results.

4. **Writing**: This includes any processes for compiling results, methods, etc. into the final paper form. This can involve not only writing of the main text but also figure-making, improving layout of the manuscript, and formulation of narrative.

   Answer: **[C]**

   Explanation: AI agents generated all figures and tables, wrote the initial draft of the main text, and revised it after receiving sparse human suggestions. Human involvement was limited to minor reference and citation format corrections such as putting the proper bibliographystyle command in the tex document. Since AI performed the majority of the writing process, with humans contributing only minimal adjustments, the work is best categorized as mostly AI, assisted by humans.

5. **Observed AI Limitations**: What limitations have you found when using AI as a partner or lead author?

   Description: In the present study, we used AI-powered multi-agent system as the lead author to achieve nearly full autonomy of data-driven plant science research from proposing hypotheses, to designing and conducting experiments, to interpreting results, and ultimately writing scientific papers. During this process, we observed three primary challenges. First, optimal and efficient representation of domain-specific knowledge base. The agent in our study is limited to literature recommended by human scientists who know key information would be learned; however, for many open questions, there wont be a such constrained search space rather the domain expert agent is expected to learn considerable domain knowledge through internet or literature research. How to effectively and accurately organize these knowledge that human researchers have contributed for centuries remains an open challenge. Second, current LLMs may not provide directly executable computer programs for customized data analysis needs. We initially allowed the MLE agent to freely develop codebase based on analysis suggestions from the Analyst agent, but experimental results showed substantial barriers to ensure the executability of the programs. The agent struggled to fully realize good suggestions. Last, more important to research fields requiring wet-lab or field experiments, the AI system is limited to current datasets for fast iteration. For instance, in several trial rounds, our agent system suggested the use of hyperspectral indices based on the success from literature. However, the AI system is currently in the digital space only and cannot receive new data streams that require new physical actions. This may prevent the system from fully realizing its potential for the scientific discovery process.

